

**Deposition of light-absorbing particles in glacier snow of the Sunderdhunga Valley, the**
**southern forefront of Central Himalaya**
Jonas Svensson[1,2], Johan Ström[3], Henri Honkanen[4], Eija Asmi[1], Nathaniel B. Dkhar[5], Shresth
Tayal[5,6], Ved P. Sharma[5,6], Rakesh Hooda[1], Matti Leppäranta[4], Hans-Werner Jacobi[2], Heikki
Lihavainen[7,1], Antti Hyvärinen[1]
1 Atmospheric Composition Research, Finnish Meteorological Institute, Helsinki, Finland
2 Université Grenoble Alpes, CNRS, IRD, INP-G, IGE, Grenoble, France
3 Department of Environmental Science, Stockholm University, Stockholm, Sweden
4 Institute for Atmospheric and Earth System Research, Faculty of Science, University of Helsinki,
Helsinki, Finland
5 The Energy and Resource Institute, (TERI), New Delhi, India
6 TERI School of Advanced Studies (TERI SAS), New Delhi, India
7 Svalbard Integrated Arctic Earth Observing System, Longyearbyen, Norway
Correspondence to: jonas.svensson@fmi.fi
Abstract
Anthropogenic activities on the Indo-Gangetic Plain emit vast amounts of light-absorbing particles
(LAP) into the atmosphere, modifying the atmospheric radiation scheme. With transport to the nearby
Himalayan mountains and deposition to its surfaces the particles contribute to glacier and snowmelt via
darkening of the highly reflective snow. The Central Himalayas have been identified as a region where
LAP are especially pronounced in glacier snow, but still remain a region where measurements of LAP
in the snow are scarce. Here we study the deposition of LAP in five snow pits sampled in 2016 (and
one from 2015) from two glaciers in the Sunderdhunga valley, state of Uttarakhand, India, Central
Himalaya. The snow pits display a distinct melt layer interleaved by younger snow above, and older
snow below. The LAP exhibit a distinct vertical distribution in these different snow layers. For the
analyzed elemental carbon (EC), the younger snow layers in the different pits show similarities, and
can be characterized by a deposition constant of about 50 µg m$^{-2}$ mm$^{-1}$ while the old snow layers also
indicate similar values, and can be described with a deposition constant of roughly 150 µg m$^{-2}$ mm$^{-1}$.
The melt layer, contrarily, display no similar trends between the pits. Instead, it is characterized by very
high amounts of LAP, and differ in orders of magnitude for concentration between the pits. The melt
layer is likely a result of strong melting that took place during the summers of 2015 and 2016. The
mineral dust fractional absorption is slightly below 50 % for the young and old snow layer, whereas in
the melt layer is the dominating light absorbing constituent, thus, highlighting the importance of dust



in the region. Our results indicate the problems with complex topography in the Himalaya, but
nonetheless, can be useful in large-scale assessments of LAP in Himalayan snow.





## 1 Introduction

Aerosol particles in the Indo-Gangetic Plain (IGP) are produced in great mass and number. Being especially prominent in the pre-monsoon season, a large fraction of the airborne aerosols are carbonaceous particles, consisting of organic carbon (OC) and black carbon (BC). Originating from the combustion of fossil fuels and biomass, the particles form the atmospheric brown cloud—known to modify the atmospheric radiation scheme (Lau et al., 2006; Menon et al., 2010; Ramanathan and Carmichael, 2008). Through air mass transport the aerosol can be conveyed and lifted from the IGP to its northern barrier, the mountains of Himalaya (e.g. Hooda et al., 2018; Kopacz et al., 2011; Raatikainen et al., 2014; Zhang et al., 2015). Covered with vast amounts of snow and ice, the Himalayan cryosphere is affected by the deposition of carbonaceous aerosol onto its surface (e.g. He et al., 2018; Jacobi et al., 2015; Ménégoz et al., 2014; Xu et al., 2009). This is due to the particulates and especially BC effectiveness in reducing the snow albedo (Warren and Wiscombe, 1980), which ultimately leads to accelerated snow melt (Flanner et al., 2007; Jacobi et al., 2015; Jacobson, 2004; Ming et al., 2012).

In addition to BC and OC, other particles such as mineral dust (MD) and snow microbes (collectively known as light-absorbing particles (LAP)) are also of importance in reducing snow albedo (e.g. Skiles et al. 2018). In Himalayan snow and ice, the LAP content has been shown to vary significantly, both spatially and temporally (e.g. see review by Gertler et al., 2016). Further, an extensive compilation of BC measurements in snow over the Tibetan Plateau is presented in the supplement of He et al. (2018), with concentrations ranging from 1 to 3600 ppb in the region termed as Himalaya. In addition to long range transported LAP, local sources within the Tibetan plateau have also been documented to be significant in some regions (e.g. Li et al., 2016), creating several different sources of LAP in the snow. Varying meteorology and terrain induced exchange processes (advection and turbulence) in the mountains further complicates the interplay between the atmospheric deposition of LAP and the snow surfaces.

Observations are further supported by modeling studies, which indicate certain sub-regions of the Himalaya to be especially vulnerable to LAP deposition. Santra et al. (2019) recently simulated the BC impact on snow albedo and glacier runoff in the Hindu Kush-Himalaya region. The authors identified a hot-spot zone for BC in the vicinity of Manora peak, located in the Indian state of Uttarakhand, central Himalaya (also sometimes called western Himalaya depending on classification). The BC induced a greater albedo reduction on glacier snow in the vicinity of this hot spot area compared to other areas in the Hindu Kush-Himalayan area. Similarly, another modeling study simulated the impact of LAP on High Mountain Asia snow albedo and its associated forcing and identified the same general area as a region where snow is especially affected by LAP-caused snow darkening (Sarangi et al., 2019). Both





of these studies (as well as the work of He et al., 2018) emphasized the need for more *in situ*
measurements of LAP in the snow of this region of the Himalaya.

Previously, we reported in Svensson et al. (2018) the measured LAP concentrations and properties in
the snow from two glaciers in the Sunderdhunga valley, located in Uttarakhand, India, central Himalaya.
While we mainly focused on the surface snow layer and characterizing the LAP, results from one 1.2
m deep snow pit were also presented. Based on the LAP concentration profile and pit stratigraphy, the
pit was estimated to represent 5 seasons. Newly sampled snow pits have since then been analyzed from
the same two glaciers, along with available automatic weather station (AWS) data from the same valley.
Here we revisit the previous interpretation of the published pit (in Svensson et al., 2018), and report the
results of our newly sampled snow pits. By comparing the BC profiles among 6 pits we aim at
quantifying the deposition of elemental carbon (EC; used here as a proxy for BC) in this area of the
Himalaya. In addition, we explore the relative contribution of MD to LAP in the different pits.

## 2 Methodology
2.1 Glaciers snow sampling and filtration
Snow was collected on Bhanolti and Durga Kot glaciers during a field campaign in the Sunderdhunga
valley (located in the Bageshwar district) in October of 2016. The two glaciers are positioned adjacent
to each other in a general northeast-southwest orientation (cf. Fig. 1) on the southern fringe of the
Himalayan mountain range and are further described in Svensson et al. (2018). Local emissions of
carbonaceous aerosol in the Sunderdhunga valley are very limited. The valley is not accessible by car
and the glaciers are at a three to four-day hike from the nearest road. On route to the glaciers the last
settlement is Jatoli, located in a river valley at an elevation of 2400 m. a.s.l. about 10 km southeast in a
perpendicular orientation to the glacier valley. Biomass burning is a common practice for cooking and
heating in Jatoli, thus some emissions from the village may enter the glacier valley. It is expected,
however, that the majority of carbonaceous particles in the glacier valley originates from regional and
long-distance transport. The relatively low elevation span as well as the glaciers' position on the
southern slopes of the Himalayan mountains nonetheless, make them more prone to LAP deposition
compared to other glaciers in the Himalaya and Tibetan plateau. Previous studies have reported elevated
LAP content in lower elevation snow for Himalayan glaciers (e.g. Ming et al., 2013), and higher
concentrations of LAPs in glaciers on the southern edge of the Himalaya (e.g. Xu et al., 2009).

On Durga Kot glacier two snow pits (hereafter Pit A and B; Fig. 1) were dug in the vicinity of each
other (~20 m) in an reachable area of the percolation zone of the glacier. Bhanolti glacier was more
easily accessible, and the three excavated snow pits (hereafter Pit C, D, E; Fig. 1) were spread out over
a greater distance (~500 m) on the glacier (see table 1 and Fig. 1 for additional information). The depth



of the pits depended on the level at which a hard layer was found, and digging could not be further
conducted. The deepest snow pit that was analyzed previously in Svensson et al. (2018), referred to as
pit 5 in that study, is from Bhanolti glacier in September of 2015, and we denote as Pit F in the
subsequent sections of this manuscript. As for the other pits from 2016, the depth of Pit F was governed
by the depth at which the hard layer was encountered.

Three distinctly different colored snow layers could be observed repeating in all but one of the year
2016 pits: a relatively thin (on the order of centimeters) very dark layer was separated by white snow
above and more grey appearing snow below. Due to this stratigraphy, we hereafter simply refer to the
whitest snow as young snow, the darkest layer as the melt layer, and the grey snow as old snow.
Representative samples ranging from 3 to 10 cm thick layers were taken throughout each pit for analysis
of LAP. Snow density measurements were conducted with a snow density kit in the upper part of the
pits (in 5 cm increments) by weighing the known volume of the sampler filled with snow. The observed
densities ranged between 0.29 and 0.46 g cm$^{-3}$ (see table 1 for details). Density measurements were not
possible below the melt layer due to the hard snow. For these layers the density was assumed 0.5 g cm$^{-3}$
(to represent aged snow) in our further analyzes. Snow density measurements were not conducted for
Pit F, and we assigned a density of 0.35 g cm$^{-3}$ for the top layer (0-3 cm; similar to observations made
in 2016), followed by 0.4 g cm$^{-3}$ between 3-10 cm depth, and 0.5 g cm$^{-3}$ for all layers below 10 cm.
Since the snow samples could not be transported in a solid phase back to the laboratory, they were
melted and filtered at the nearby base camp using the same principles as in Svensson et al. (2018).
Filters were transported back to the analysis laboratory in petri slides.

2.2 Meteorological observations
In September 2015 an AWS was installed next to the glacier ablation zone of Durga Kot (Fig. 1) about
1.5 km northwards at an elevation below the snow sampling sites. The AWS is equipped with
instruments for air temperature, relative humidity, shortwave (SW) and longwave (LW) radiation (up
and down), wind speed and direction, and snow depth (Campbell Scientific SR50 Ultrasonic Distance
Sensor). In this paper we use the snow depth data between September 2015 and September 2017 to
estimate the local precipitation. The original snow depth data, logged once every 10 minutes was filtered
to daily resolution by applying a moving median window of 24 hours and for the noon value of each
day in further analyzes. This filtering removed much of the signal noise. However, before this filtering
was applied the data was reduced using several logical conditions such as: the incoming SW radiation
is greater than outgoing SW radiation (to remove errors due to sensors covered by snow), and the surface
albedo is greater than 0.2 (to ensure snow cover). Finally, the consistency between the daily albedo and
snow depth was inspected using data presented in Figure S1a. Each day the snow depth increased was
interpreted as precipitation, and to arrive at an estimate of the snow water equivalent (SWE), the fresh



snow density is assumed to be 100 kg m$^{-3}$. The solid precipitation derived based on the cumulative SWE
is presented in Figure S1b.

2.3 Filter analysis
The analysis of filters followed the procedure in Svensson et al. (2018), with transmission
measurements coupled with thermal-optical analysis. According to the measurement nomenclature
(Petzold et al., 2013), the carbonaceous constituents measured are EC and OC. The measurement
method briefly follows the procedure of placing a filter punch in a custom-built particle soot absorption
photometer (PSAP) to measure the transmittance (at $\lambda$ = 526 nm; Krecl et al., 2007)—providing an
optical depth for all of the particles captured by the filter. The filter punch is then placed in an OCEC
analyzer (Sunset instrument, using the EUSAAR_2 protocol) to determine the OC and EC mass,
followed by another measurement with the PSAP. The OCEC analysis removes the carbonaceous
species and, thus, by comparing the PSAP results obtained before and after the analysis, the relative
contribution of the light absorption by EC particles in the total particles optical depth is obtained. The
remaining optical depth we attribute as non-EC material. This fraction of the total optical thickness we
report as the percentage of the mineral dust absorption on the filter samples (expressed as $f_D$). For further
details concerning the measurements see Svensson et al (2018).

Some of the filter samples (N=17) were saturated with too much light absorbing material prohibiting
reliable EC measurements despite reducing the sample to a melted equivalent of only 30 mL. To
mitigate this problem, we calculated the EC indirectly from the analyzed total carbon (TC) for the
saturated samples. From OCEC analysis TC is the most robust measured constituent, since it includes
both OC and EC and is not affected by their split point, which may be incorrectly placed for very dark
filters (Chow et al., 2001). A linear relation was fitted for non-saturated filter samples and the obtained
correlation of $EC = 0.10TC + 0.12$ was used to reconstruct the EC content for the filter samples
containing high amounts of absorbing particles (see details in supplement and Fig. S3a-b). The slope
compares well with the slopes reported for air samples collected at two sites in the Himalayas about
550 km south-east from Sunderdhunga in the Kathmandu valley 32 km (altitude of 2150 m a.s.l.) east
of Kathmandu, and Langtang 60 km north of Kathmandu (altitude of 3920 m a.s.l) (Caricco et al., 2003).
There, the authors found that the EC/TC ratio was 0.17 for both sites during the summer monsoon
season, but between 0.10 and 0.13 during what they described as the ramp-up period and the peak
concentration season. The snow samples do not have an upper limit for particles sizes, whereas the air
samples were collected as PM2.5 (particulate matter collected below an aerodynamic diameter of 2.5
µm). The slopes are rather similar to our value, and the authors found as well a very strong correlation
of 0.89 (r$^2$) between monthly average EC and OC.



## 3 Results and discussion

### 3.1 EC deposition in young and old snow samples

When the EC content is analyzed from filtered snow samples, a common practice is to convert the results into mass concentrations [EC], given per volume or mass of melt water (e.g. µg L$^{-1}$ or ng g$^{-1}$). A spread in results is often largely due to local processes and specific sampling layer thicknesses. The mass deposition per unit area $\widetilde{EC}$, on the other hand, can be expected to be less variable with increasing number of layers used to calculate this value. The deposition in each layer is calculated according to:

$$\widetilde{EC_i} = [EC]_i \frac{\rho_{s_i}}{\rho_w} d_i \tag{1}$$

were $\rho_s$ and $\rho_w$ are snow and liquid water densities, respectively. The index $i$, is the number of the sampled layer from top to bottom, and $\rho_s/\rho_w\, d$ is the SWE thickness, $d_{SWE}$. The $\widetilde{EC_i}$ and $d_{SWEi}$ are transformed to cumulative plots by integrating over the layers from the surface to the bottom. These profiles are presented in Fig. 2a-f (with each sampling layer represented by a square).

The visible snow pit stratigraphy described above in section 2.1 can be observed in the pit profiles. At the top, the accumulated EC ($ECacc$) as a function of the accumulated $d_{SWE}$ ($SWEacc$) portray the young snow layers, whereas in the bottom of the pits the data points represent the old snow layers (Fig. 2a-f). This pattern (with both young and old snow layers) is visible in pits A, B, C, and D (Fig. 2a-d). These pits also have the melt layer interleaved between the young and old snow layers, indicated by the sharp increase (or steep slope) between the young and old snow layers. In the two pits where this general outline is not visible (pit E Fig. 2e and pit F Fig. 2f), it can be explained by the fact that pit E extended only to the melt layer (therefore no old snow samples) while pit F had essentially no young snow samples at the time of sampling (therefore pit F starts with the melt layer).

With the data points for young and old snow appearing rather similar in slope between the pits, the homogeneity is emphasized further by comparing the observations with common effective constants for young and old snow ($EC_y^*$ and $EC_o^*$), respectively. Suitable constants were determined to be close to 50 µg m$^{-2}$ per mm SWE for young snow and 150 µg m$^{-2}$ per mm SWE for old snow (see supplement). The resulting deposition using $EC_y^*$ and $EC_o^*$ are superimposed over the observations in Figure 2a-e as dashed lines for young snow and dash-dotted lines for old snow. These lines then represent the constant deposition of EC as function of accumulated melt water in a column according to:

$$ECacc = constant * SWEacc + offset \tag{2}$$



where ECacc is the accumulated EC mass per m$^2$ and SWEacc is the accumulated melt water in L m$^{-2}$
(or mm), and the 'constant' is the deposition constant. The offsets for young snow are a result of
enhanced observed EC concentration in the top layer, which can numerically be compensated for by
"artificially" adding a small value to (ΔSWEacc) to each pit, which in essence dilute the top layer, but
have marginal effect on the overall picture. This meant simply rewriting the linear relation above into:

222                    $ECacc = constant * (SWEacc + \Delta SWEacc)$                    (3)


The ΔSWEacc amounts were chosen by trial and error to be in multiples of 10 mm for simplicity. The
resulting values were 10, 10, 60, 20, and 20 mm for pits A through E in order to explain the apparent
offset. A physical interpretation of these numbers may be the loss of water from the surface layer due
to evaporation or sublimation, which enhance $[EC]$ in the top layer. For the old snow layers, snow and
EC were numerically removed in the data by subtracting accumulated EC and SWE (including the melt
layer, when present) down to the old snow layer.  This was done such that the first data point satisfies
$EC_o^*$. Hence, for old snow $[EC]_1 d_{SWE_1}/EC_o^* = SWE_{acc_1}$ where the index (1) represents the top layer of
old snow.

By applying the offset values and numerically removing the upper snow layers, we compare the data in
Fig. 2a-f in two separate figures (Fig. 3a-b), one where young snow are grouped together and one for
old snow. In Fig 3a, the observed $ECacc$ is plotted against the $ECacc$ value if $EC_y^*$ is used. In Fig. 3b
the observed $ECacc$ is plotted against the $ECacc$ value if $EC_o^*$ is used. Note that for old snow the first
data point in the different pits will, by definition, be on the 1:1 line. Nevertheless, the consistency
between the pits is striking and the fact that much of the variation in $ECacc$ as function of $SWEacc$ (or
depth in the pit) can be explained by $EC_y^*$ and $EC_o^*$ alone is a very interesting finding.

3.2 Melt layer
On the contrary to the observed similarities in the different pits between young and old snow, the melt
layer samples do not display similar trends. Instead of being characterized by a common constant, the
$ECacc$ value as function of $SWEacc$ in the melt layer differs by orders of magnitude between the
different pit profiles. To explore the melt layers further, we make use of the constant for young snow,
$EC_y^*$. Assuming that this is a characteristic value for precipitation during the winter season, we can
estimate the required amount of precipitation ($SWEacc$) that is needed to explain the observed $ECacc$
deposition. These derived precipitation amounts for each pit are presented in Figure 4 as a function of
the relative depth from the surface to the bottom of the pit. Using this approach, pit F corresponds to a
total equivalent of about 2100 mm in precipitation, whereas pits B, E, and D represent 3500, 4300, and
5100 mm, respectively. Pits A and C deviate starkly from the others, with 37000 and 55000 mm





precipitation. Comparing these derived values to other precipitation estimates allows us to provide a
temporal perspective required to explain the observed EC in the pits. Other studies have shown that the
annual precipitation is very altitude-level dependent in the Himalayas, and based on the altitude of the
glaciers alone one would expect less than about 1000 mm in annual precipitation (Anders et al., 2006;
Bookhagen and Burbank, 2010). Based on the changes in snow depth, the local precipitation was
estimated using the AWS as described in section 2.2. This analysis gave a snow accumulation of about
600 mm SWE in the winter season 2015-2016 and 700 mm in the 2016-2017 winter season at the
location of the AWS. Over the season, a fraction of the snow evaporates or sublimates, possibly
accounting for a magnitude of mm per day during favorable conditions (Stigter et al., 2018). Further,
Mimeau et al. (2019) estimated the sublimation between 12 and 15 % of the total annual precipitation
in the Khumbu valley, Nepal. This amount might be missed by this method using daily data.
Nonetheless, our two precipitation estimates are below the observed annual precipitation of 976 mm in
2012/2013 at 3950 m altitude, about 250 km to the north-west next to the Chhota Shigri glacier front
(Azam et al., 2016). Measured with an automatic precipitation gauge (i.e. capturing all precipitation
forms), the authors found that the majority of precipitation was during the winter season, and that the
summer monsoon contributed with only 12 % to the annual precipitation. Based on these observation
estimates, and the similarities with our Sunderdhunga AWS precipitation patterns, we estimate
that about $800 \pm 200$ mm is a characteristic annual precipitation amount close to where the pits were
dug. If the precipitation amounts derived to explain the deposited EC in each pit is divided by 800 mm,
the minimum number of years required to explain the EC observed in the pit is acquired. With this
approach it is clear that it would require decades of precipitation to explain the EC in the melt layers in
pits A and C. This is unrealistic, especially when the lower levels in pit F from the previous year is
compared. Even the difference in EC amount between pits B, E, and D compared to F is much greater
than can be explained from aggregating the EC accumulated by one year of precipitation in a single
melt layer. This leads us to propose that EC must have been transported laterally in the surface layer
during the melt period in the summer of 2016 and converged in the altitude range where the pits were
dug. From Figure 1 it can be seen that the pits were dug in a complex terrain where slopes with
increasing gradient are reaching up to the summit towards the southwest.

The data and analysis presented above lead us to propose that the old snow layers observed in pit F
from 2015 are the same old snow layers observed for the pits dug in 2016. The EC equivalent
precipitation profile of pit F presented in Figure 4 suggests that strong melting had taken place already
in summer 2015. Hence, the old snow is composed of snow from at least the season 2013-2014 (or
perhaps also earlier seasons). Stratigraphy analysis for pit F presented in Svensson et al. (2018)
suggested that the snow deposition represented five seasons. The amount of precipitation represented
by the EC deposition (cf. Figure 4) in the old snow is about 2100 mm, which suggests that the EC was
deposited over several seasons, but less than 5 seasons. Another strong melt took place in 2016, possibly



leading to melting all of the snow from the season 2015-2016. In addition, during the melting phase,
water and snow particulates could be transported down the slopes from areas of the glacier with steep
slopes. Because the steepness of the slope decreases towards the valley, this resulted in a convergence
of percolated material from areas above the sampling sites. The young snow is likely part of the 2016-
2017 winter season that had started to accumulate before the sampling in October 2016 was
commenced. This is confirmed by AWS data that indicates intermittent snow events in October 2016.
At the AWS location a seasonal snow cover was in place in December 2016.

3.3 Mineral dust fraction in snow
An initial inspection of the mineral fractional absorption on the filters did not reveal any special
common pattern in concentration between the different pits, except for the melt layer samples, which
appeared to have higher concentrations than the other samples. In Figure 5, the data is grouped
according to the pit stratigraphy classification, and although the absolute range of MD fractions in
young snow samples is very large (5 to 71 %), the quartile range is only between 32 to 48 % with a
median value of 39 %. The median value for old snow is somewhat larger at 46 %, along with the range
and quartiles, which are closer together, from 26 to 70 % and from 43 to 50 %, respectively. The range
of values for the melt layer are consistently higher compared to the other two snow types. The median
is 78 % with a range and quartiles of 48 to 95 % and 74 to 82 %, respectively. Note that from a total of
95 samples only 16 are from the melt layer.

Due to the typically heavy loading of material on the filters obtained in the melt layer, those values
should be taken with caution. Non-linear effects could skew the resulting light absorption fractions
towards larger values science with a very heavy loading (dark filter) the contribution by remaining
particles may be over-estimated. This is because the relative contribution by additional light absorbing
material decreases as the amount of material increases on very dark filters. In an extreme case, black on
black will not add any contribution. The larger range of values in young snow compared to old snow is
possibly an effect from the geometric thickness of the sampled slabs, which are in young snow generally
thinner than in old snow, and that the density of young snow is typically less than the density of old
snow. This results in each of the sampled segments in young snow representing less deposition of both
water and LAP and, therefore, presenting a larger variability. Nevertheless, the ensemble of data
presents similar median values for both young and old snow. The median of the percentage of the
mineral dust absorption $f_D$ value for young and old snow samples together becomes 44 %. The specific
absorption by minerals is expected to be orders of magnitude smaller than BC (e.g. Utry et al., 2015),
and the same is expected with respect to EC. This suggests that the deposition of minerals in the snow
is orders of magnitude larger than EC. If we simply scale our characteristic EC constants ($EC_y^*$ and





$EC_o^*$), with the median of $f_D$ and the ratio between their specific mass absorption coefficients (MAC),
according to:

$$\frac{f_D}{(1-f_D)} \frac{MAC_{EC}}{MAC_D} EC_c = D_c \qquad (4)$$


we arrive at a mass concentration for minerals. We use a MAC for BC of 7.5 m$^2$ g$^{-1}$ (Bond and
Bergstrom, 2006). The MAC for the minerals is not known and can vary significantly, but for the sake
of this test we use a MAC value representative for the mineral quartz with 0.0023 m$^2$ g$^{-1}$ (Utry et al.,
2015). If we use these values we arrive at a range of 128-384 µg g$^{-1}$ of minerals in the snow. This is in
range with previous gravimetric observations from Himalaya (e.g. Thind et al., 2019; Zhang et al.,

334  2018).


3.4 Discussion
Our results indicate that the contribution to light absorption by minerals can be comparable to light
absorption by EC in the Sunderdhunga area at about 5 km altitude. This translates into a mass
concentration ratio between EC and minerals of more than three orders of magnitude. These large ratios
are typically not reported for air samples because much of the deposited minerals are likely from local
sources. This supports a hypothesis of a positive climate feedback that results in a reduction of snow
cover and the exposure to larger sources of minerals.

For the Tibetan plateau, Zhang et al. (2018) estimated that the retreat of the snow cover could be
advanced by more than a week due to LAP in snow. In their estimates, BC accounted for most of this
effect and dust advanced the melting by about one day. The BC concentration in snow used in their
calculations were about one order of magnitude larger than our derived values form the profiles in the
snow pits. This difference can be attributed to the significant contribution of aerosol particle dry
deposition in arid regions (Wang et al., 2014), but the range of values presented in their Table 2 reveals
a potential problem from sampling surface snow. Post depositional processes (e.g.
sublimation/evaporation, hoar formation, snow drift) can alter the concentration at a given location
relatively fast, which is less of a problem if a deeper layer of the snow pack is investigated instead of
solely the surface snow. Simply taking a larger vertical slab is not sufficient as is evident from the melt
layer in the present study. The melt layer in the pits can be studied to characterize the short-term
seasonal surface albedo, but the aerosol concentrations cannot be directly related to the deposition. The
consistency between pits and different sampling seasons in the integrated deposition profiles above and
below the melt layer show the strength in the data collected from snow pits in comparison to snap-shot
conditions of surface snow.



## 4 Conclusions

In this study we aimed at characterizing the observed deposition of EC in the glacier snow in the Sunderdhunga valley and to estimate the contribution from minerals to LAP in the snow. The analysis illustrates that in the sampling area of Durga Kot and Bhanolti glaciers, the deposition of EC in young snow (from current winter season) is characterized by approximately $50\,\mu g\,m^{-2}\,mm^{-1}$ SWE water, which is in the range of other observations. The median fraction of light absorption caused by minerals was about 39 % (Q1=32, Q3=48). In old snow (from previous winter seasons), the deposition was characterized by about $150\,\mu g\,m^{-2}\,mm^{-1}$ SWE water. The reason for this difference can simply be due to a larger deposition in the years before sampling was conducted, or that more water had the chance to leave the snow-pack of older snow. Different from young snow, old snow have had to survive at least one summer season. The median fraction of light absorption was 46 % (Q1=43, Q3=50) by minerals in the old snow layer. Although the variability within each layer is rather large, the obtained lower median fraction for young snow is consistent with the fact that old snow is more exposed to rock surfaces free of snow during the summer season.

Between these two layers of old and young snow, a clearly visible and very dark layer was present. This layer was most likely a result of strong melting that took place in the summers of 2015 and 2016. However, the high concentration of EC found in this layer cannot simply be explained by a collapse of the snow-pack vertically, and thus it is concluded that lateral transport of LAP (including EC and minerals) took place that resulted in a convergence of material in the altitude range of the snow pits. Different from the other two layers (young and old snow), this melt layer presented large differences with respect to EC content among the different pits. The fraction of light absorption by minerals was the highest of the three layers and was about 80 % (Q1=74, Q3=82).

The profiles of EC and the mineral absorption fraction show good agreement between subsequent years and among different pits. At the same time, the topography in this mountainous region of Himalaya evidently causes great complexity with respect to the distribution of LAP in the snow surface layer during periods of strong melt. Although data is limited in spatial and temporal dimensions our results are useful for large scale radiation impact assessments of EC deposition and minerals. In small scale regional studies, however, the effects of complex topography and spatial variability should be considered separately. Future work should further study the mineral dust and its composition in the area, in order to more accurately elucidate dust role in the snow radiation scheme in this part of the Himalaya.



Data availability

All data are available upon request.

Author contributions

J. Sv, H.H, E.A., N.D., H.L., participated in the field expedition. S.T., R.H., V.S., M. L., H.L., A.H. handled project administration. Data analysis was performed by J. Sv. and J. St. Funding acquisition: A.H. Supervision M.L. and H.L. J. Sv led the writing of the manuscript with J. St., with input from all other co-authors.

Completing interests.

The authors declare that they have no conflict of interest.

Acknowledgements

This work has been supported by the Academy of Finland project: Absorbing Aerosols and Fate of Indian Glaciers (AAFIG; project number 268004), and the Academy of Finland consortium: "Novel Assessment of Black Carbon in the Eurasian Arctic: From Historical Concentrations and Sources to Future Climate Impacts" (NABCEA project number 296302). J.Svensson acknowledges support from the two Finnish foundations: Maj and Tor Nessling and Oskar Huttunen; as well as the invited scientist grant from the UGA. J. Ström is part of the Bolin Centre for Climate Research, and acknowledges the Swedish Research Council grant 2017-03758. We are thankful for Daniela Tuomala's work with the filter analyzes, as well as the strenuous assistance given by Sherpas and mountain guides during the expeditions to the Sunderdhunga valley.



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






Table 1. Snow pit details from Sunderdhunga valley. Durga Kot glacier snow pits are A-B, while C-F are from Bhanolti glacier.

| Snow pit ID and elevation (m a.s.l) | Depth interval (cm) | Snow density (g cm⁻³) Measured | Snow density (g cm⁻³) Assumed | Water equivalent (mm m⁻²) | TC analyzed (µg L⁻¹) | EC (µg L⁻¹) Analyzed | EC (µg L⁻¹) Reconstructed | EC deposition (µg m⁻²) | fD (%) |
|---|---|---|---|---|---|---|---|---|---|
| A, 5055 | 0-3 | 0.38 | | 11.4 | 1130 | - | 120 | 1364 | 24.6 |
| | 3-6 | 0.38 | | 11.4 | 238 | 18 | | 207 | 29.2 |
| | 6-9 | 0.35 | | 10.5 | 477 | 47 | - | 495 | 40.4 |
| | 9-12 | 0.37 | | 11.1 | 30300 | - | 3125 | 34688 | - |
| | 12-15 | 0.39 | | 11.7 | 1307404 | - | 134685 | 1575819 | 76.1 |
| | 15-20 | | 0.50 | 25.0 | 68177 | - | 7034 | 175855 | 55.1 |
| | 20-25 | | 0.50 | 25.0 | 1398 | 278 | - | 6945 | 47.9 |
| | 25-30 | | 0.50 | 25.0 | 1549 | 147 | - | 3684 | 49.8 |
| | 30-35 | | 0.50 | 25.0 | 1769 | 271 | - | 6787 | 41.9 |
| | 35-40 | | 0.50 | 25.0 | 1466 | 251 | - | 6273 | 46.5 |
| | 40-45 | | 0.50 | 25.0 | 883 | 141 | - | 3528 | 44.6 |
| | 45-50 | | 0.50 | 25.0 | 751 | 142 | - | 3553 | 43.1 |
| | 50-60 | | 0.50 | 50.0 | 1090 | 171 | - | 8544 | 51.5 |
| | 60-70 | | 0.50 | 50.0 | 763 | 88 | - | 4412 | 45.9 |
| B, 5055 | 0-3 | 0.40 | | 12.0 | 1542 | 95 | - | 1143 | 38.3 |
| | 3-6 | 0.40 | | 12.0 | 693 | 30 | - | 364 | 27.5 |
| | 6-9 | 0.39 | | 11.6 | 31710 | - | 3291 | 38015 | 77.8 |
| | 9-12 | 0.33 | | 9.9 | 69667 | - | 7210 | 71378 | 75.0 |
| | 12-15 | 0.33 | | 9.9 | 3498 | - | 374 | 3699 | 50.6 |
| | 15-19 | | 0.50 | 20.0 | - | - | 267 | 5348 | 49.9 |
| | 19-29 | | 0.50 | 50.0 | 1534 | 246 | - | 12319 | 49.8 |
| | 29-39 | | 0.50 | 50.0 | 1295 | 190 | - | 9480 | 46.2 |
| | 39-49 | | 0.50 | 50.0 | 1517 | 248 | - | 12407 | 52.1 |





| | | | | | | | |
|---|---|---|---|---|---|---|---|
| 49-59 | | 0.50 | 50.0 | 1753 | 182 | - | 9100 | 40.2 |
| 59-69 | | 0.50 | 50.0 | 733 | 103 | - | 5156 | 41.2 |
| 69-79 | | 0.50 | 50.0 | 730 | 102 | - | 5121 | 44.9 |
| **C, 5068** | | | | | | | | |
| 0-3 | 0.40 | | 12.0 | 2386 | - | 249 | 2983 | 47.6 |
| 3-6 | 0.39 | | 11.7 | 590 | 45 | - | 523 | 31.6 |
| 6-11 | 0.39 | | 19.5 | 372 | 34 | - | 658 | 59.4 |
| 11-16 | 0.42 | | 21.0 | 799 | 93 | - | 1959 | 54.8 |
| 16-21 | 0.46 | | 23.0 | 1074 | 141 | - | 3240 | 58.0 |
| 21-26 | | 0.50 | 25.0 | 1047065 | - | 107865 | 2696629 | - |
| 26-31 | | 0.50 | 25.0 | 4480 | 370 | - | 9257 | 62.0 |
| 31-36 | | 0.50 | 25.0 | 684 | 80 | - | 1988 | 58.9 |
| 36-41 | | 0.50 | 25.0 | 906 | 150 | - | 3746 | 43.6 |
| 41-46 | | 0.50 | 25.0 | 658 | 126 | - | 3159 | 44.2 |
| 46-56 | | 0.50 | 50.0 | 863 | 137 | - | 6871 | 43.5 |
| 56-66 | | 0.50 | 50.0 | 1191 | 156 | - | 7803 | 45.7 |
| 66-76 | | 0.50 | 50.0 | 832 | 144 | - | 7222 | 44.9 |
| 76-86 | | 0.50 | 50.0 | 802 | 94 | - | 4709 | 45.9 |
| 86-96 | | 0.50 | 50.0 | 416 | 51 | - | 2543 | 42.6 |
| 96-106 | | 0.50 | 50.0 | 609 | 78 | - | 3913 | 45.3 |
| 106-116 | | 0.50 | 50.0 | 692 | 76 | - | 3821 | 50.0 |
| 116-126 | | 0.50 | 50.0 | 500 | 46 | - | 2322 | 57.9 |
| 126-136 | | 0.50 | 50.0 | 1265 | 108 | - | 5386 | 59.0 |
| **D, 5125** | | | | | | | | |
| 0-3 | 0.39 | | 11.7 | 1135 | 127 | - | 1487 | 35.2 |
| 3-6 | 0.39 | | 11.7 | 1012 | 91 | - | 1068 | 34.0 |
| 6-9 | 0.37 | | 11.1 | 449 | 30 | - | 337 | 42.6 |
| 9-12 | 0.37 | | 11.1 | 810 | 41 | - | 450 | 47.3 |
| 12-15 | 0.37 | | 11.0 | 1089 | 84 | - | 916 | 48.5 |
| 15-18 | 0.37 | | 11.0 | 357 | 32 | - | 353 | 38.6 |
| 18-21 | 0.36 | | 10.8 | 918 | 59 | - | 637 | 38.5 |
| 21-24 | 0.42 | | 12.6 | 274 | | 36 | 448 | 70.8 |
| 24-27 | 0.42 | | 12.6 | 322 | 23 | - | 293 | 57.2 |



| Group | Bin | | | | | | | | |
|---|---|---|---|---|---|---|---|---|---|
| | 27-30 | | 0.36 | 10.8 | 443 | 28 | - | 297 | 36.3 |
| | 33-36 | | 0.36 | 10.8 | 2393 | | 253 | 2734 | 95.1 |
| | 36-39 | | 0.45 | 13.5 | 1714 | | 186 | 2506 | 77.6 |
| | 39-42 | | 0.45 | 13.5 | 6806 | | 710 | 9591 | 77.1 |
| | 42-44 | 0.50 | | 10.0 | 177424 | | 18313 | 183125 | - |
| | 44-49 | 0.50 | | 20.0 | 9733 | | 1025 | 20504 | 60.1 |
| | 49-54 | 0.50 | | 25.0 | 5708 | 665 | - | 16635 | 59.5 |
| | 54-59 | 0.50 | | 25.0 | 1743 | 232 | - | 5798 | 69.6 |
| | 59-69 | 0.50 | | 50.0 | 901 | 129 | - | 6459 | 46.1 |
| E, 5143 | 0-3 | | 0.33 | 9.9 | 992 | 128 | - | 1268 | 35.9 |
| | 3-6 | | 0.33 | 9.9 | 422 | 63 | - | 622 | 41.4 |
| | 6-9 | | 0.37 | 11.1 | 891 | 81 | - | 903 | 25.9 |
| | 9-12 | | 0.31 | 9.3 | 569 | 41 | - | 380 | 43.0 |
| | 12-15 | | 0.31 | 9.3 | 806 | 73 | - | 681 | 27.7 |
| | 15-18 | | 0.29 | 8.7 | 750 | 35 | - | 302 | 41.0 |
| | 18-21 | | 0.29 | 8.7 | 345 | 22 | - | 193 | 55.6 |
| | 21-24 | | 0.39 | 11.7 | 644 | | 81 | 943 | 4.5 |
| | 24-27 | | 0.38 | 11.4 | 500 | 50 | - | 566 | 27.0 |
| | 27-30 | | 0.38 | 11.4 | 439 | 65 | - | 739 | 56.7 |
| | 30-33 | | 0.40 | 12.0 | 395 | 53 | - | 635 | 49.4 |
| | 33-36 | | 0.40 | 12.0 | 642 | 26 | - | 308 | 27.3 |
| | 36-39 | | 0.44 | 13.2 | 397 | 33 | - | 430 | 38.9 |
| | 39-42 | | 0.44 | 13.2 | 1250 | 53 | - | 705 | 34.8 |
| | 42-45 | | 0.44 | 13.2 | 1148 | 75 | - | 988 | 48.4 |
| | 45-48 | | 0.45 | 13.5 | 828 | 169 | - | 2287 | 81.1 |
| | 48-51 | | 0.45 | 13.5 | 901 | 131 | - | 1775 | 77.7 |
| | 51-54 | | 0.45 | 13.5 | 617 | 58 | - | 786 | 85.8 |
| | 54-55 | | 0.45 | 4.5 | - | - | 4694 | 21125 | 85.8 |
| | 55-60 | 0.50 | | 25.0 | 69606 | - | 7198 | 179946 | 85.8 |
| F, 5008 | 0-3 | 0.35 | | 10.5 | 4075 | 66 | | 690 | 77.9 |
| | 3-6 | 0.40 | | 12.0 | 4821 | 273 | | 3271 | 60.8 |





| | | | | | | |
|---|---|---|---|---|---|---|
| 6-10 | 0.40 | 16.0 | 17686 | 1828 | 29242 | 69.5 |
| 10-15 | 0.50 | 25.0 | 3555 | 233 | 5830 | 60.3 |
| 15-20 | 0.50 | 25.0 | 859 | 111 | 2786 | 33.1 |
| 20-30 | 0.50 | 50.0 | 1324 | 141 | 7036 | 49.2 |
| 30-40 | 0.50 | 50.0 | 807 | 106 | 5278 | 39.6 |
| 40-50 | 0.50 | 50.0 | 890 | 98 | 4907 | 36.7 |
| 50-60 | 0.50 | 50.0 | 2825 | 270 | 13484 | 49.5 |
| 60-70 | 0.50 | 50.0 | 1228 | 179 | 8965 | 39.9 |
| 70-80 | 0.50 | 50.0 | 696 | 93 | 4650 | 36.1 |
| 80-90 | 0.50 | 50.0 | 483 | 73 | 3640 | 35.8 |
| 90-100 | 0.50 | 50.0 | 1190 | 144 | 7190 | 43.9 |
| 100-110 | 0.50 | 50.0 | 652 | 79 | 3965 | 29.5 |
| 110-120 | 0.50 | 50.0 | 554 | 57 | 2846 | 25.7 |




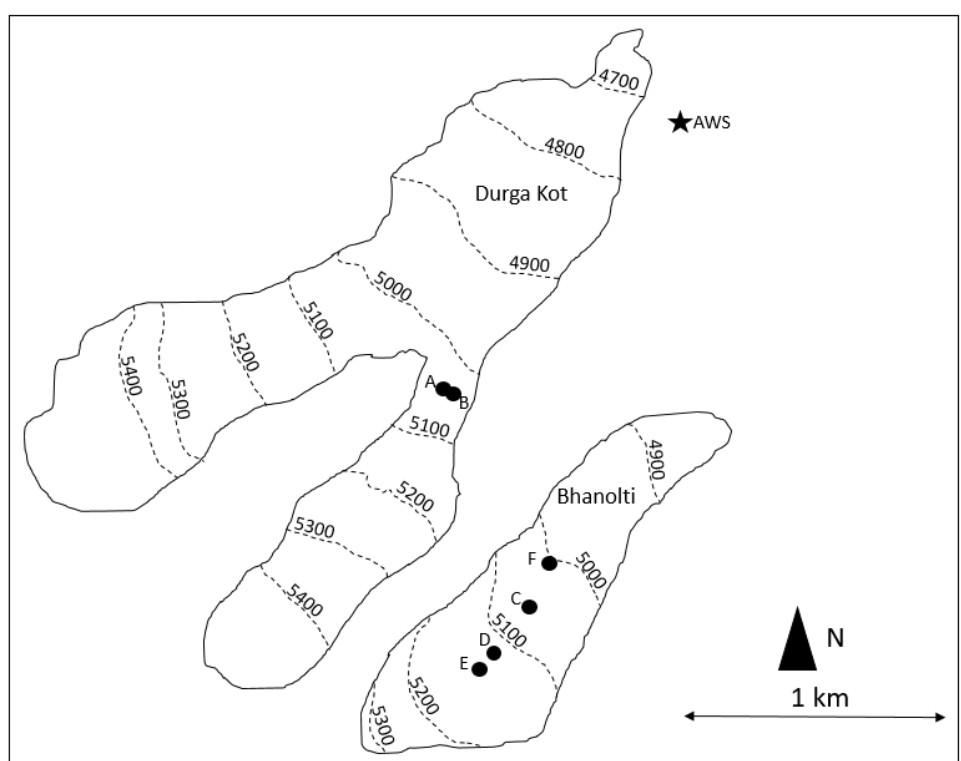


533        Figure 1. Map of glaciers with the location of the snow pits (black dots) and AWS indicated.









Figure 2. The cumulative $\widehat{EC}_i$ (ECacc) from top to bottom in the snow pits as function of accumulated $d_{SWEi}$ expressed as SWEacc (mm): (a) Pit A, (b) Pit B, (c)
Pit C, (d) Pit D, (e) Pit E, (f) Pit F. The upper dashed line represents a constant deposition $EC_y^*$ and the lower dashed-dotted line represents a constant deposition
$EC_o^*$. In pit E there were no snow samples classified as old snow, hence there is no n $EC_o^*$ line, while in in pit F there were no young snow samples, therefore no
$EC_y^*$ line.






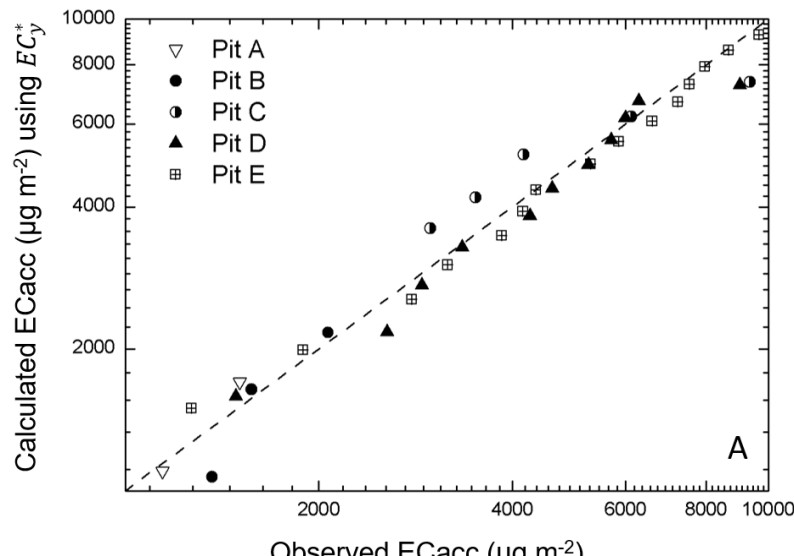


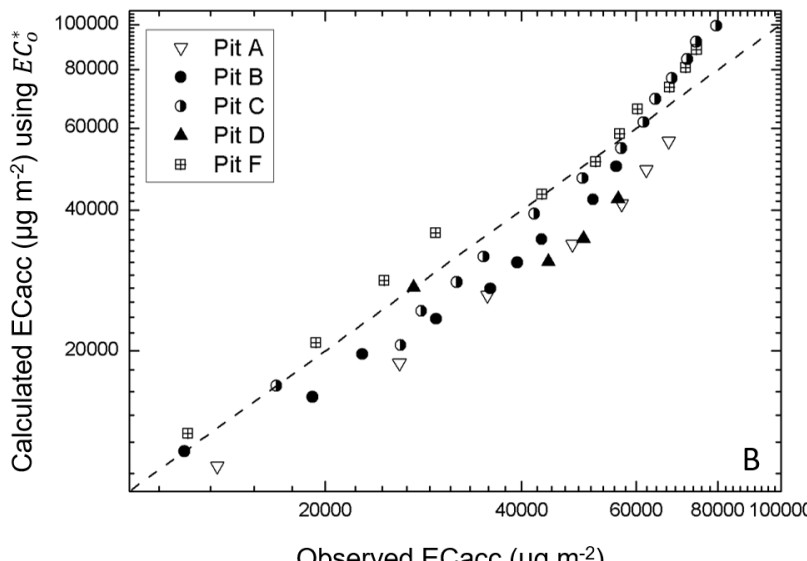


Figure 3. Observed and the calculated deposition using the constant deposition $EC_y^*$ for young (a) and $EC_o^*$ for old (b) snow samples. Dashed lines indicate a 1:1 slope.








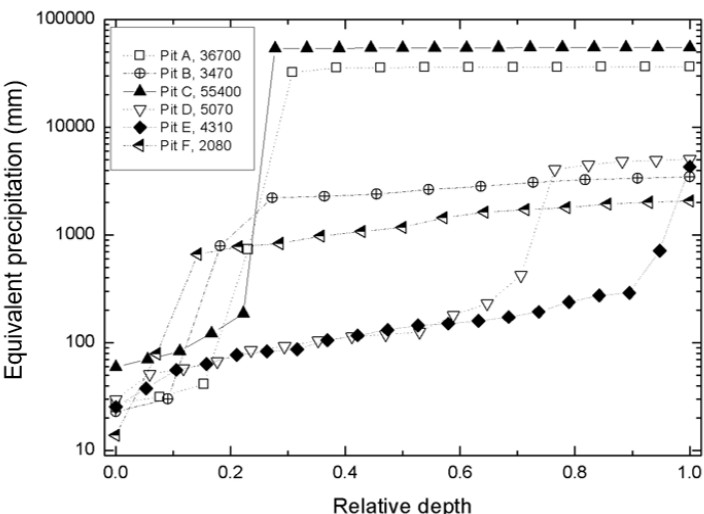


Figure 4. Equivalent precipitation for each pit based on a constant deposition $EC_y^*$ in fresh snow as
function of the relative depth of the pit from top to bottom.

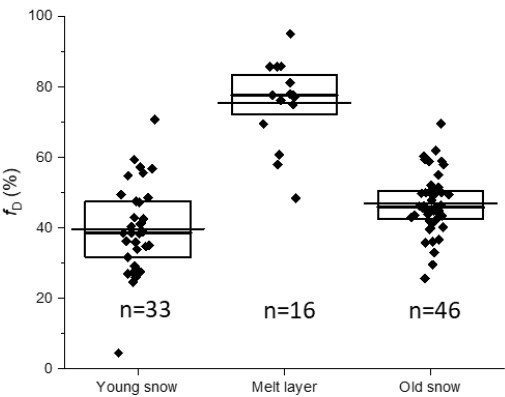


Figure 5. Fractional dust absorption remaining after burning the filters during OC/EC analysis. The
diamonds are individual values for each filter and the thin extended line represents the arithmetic
average. The box and thicker line represent the quartile range and median, respectively. The number
of samples are indicated in the figure as (n).