# Peer review of "Deposition of light-absorbing particles in glacier snow of the Sunderdhunga Valley, the"

_Atmospheric Chemistry and Physics, 2020_

## Referee Comment (RC1) · Anonymous Referee #1 · 24 Nov 2020

Remarks to the Authors

Review of "Deposition of light-absorbing particles in glacier snow of the Sunderdhunga
Valley, the southern forefront of Central Himalaya" by Jonas Svensson et al.

Manuscript Number: acp-2020-1059

—————————————————————————————————————————

General comments: This paper reports in-situ data for mass concentrations of light-
absorbing particles (LAP) in snow such as black carbon, organic carbon, and dust
collected at the Sunderdhunga Valley, the southern forefront of Central Himalaya dur-

ing in October 2016. By comparing the data with in-situ surface meteorological data collected also by the authors, mechanisms that controlled the vertical redistribution of black carbon and dust within the near-surface snow are discussed. The methods to obtain these data are reliable. So, the data themselves are valuable for the community: e.g., the data tell us the current mass concentrations of LAPs within snow in the study area; and the data can be used for evaluation of chemical transport models. However, this reviewer would like to suggest that the authors should provide clear answers to the following major concerns before considering its publication:

1. The discussion on the redistribution of LAPs in "melt layers" is not enough to me. At least, the effects of dry deposition should be examined before considering the possibility of the lateral transport of LAPs in the near-surface snow.

2. What is the main factor that induced higher concentration of MD (mineral dust) in the "melt layers"?

————————————————————————————————————————————————————

Specific comments

P. 1, L. 20: What do the authors mean by "scheme"? "Scheme" is often related to a sub-program of a numerical model. Therefore, using "state" may be better here.

P. 1, L. 28: "similarities": Can the authors describe quantitatively?

P. 3, L. 43: "scheme": same as above.

P. 3, L. 57: "ppb" -> "ppbw" or "ppbv"?

P. 3, L. 64 ∼ 65: Observations are ground truth, so that it is not necessary for observations to be supported by modelling studies. On the other hand, observations can support the reliabilities of numerical models. Please reformulate this sentence.

P. 5, L. 109: "hard layer": The "hardness" considered here may depend on the tools that the authors used to dig the pits. What kind of tools did the authors use to dig?

P. 5, L. 115 ∼ 117: Please show a figure showing profiles for snow density at these 2016 pits. Then, explain/indicate the layers where the dark, white, and gray snows can be found.

P. 5, L. 116 ∼ 117: "a relatively thin (on the order of centimeters) very dark layer was separated by white snow above and more grey appearing snow below.": Is this sentence OK? Please check it again.

P. 5, L. 118: Strictly speaking, "melt layer" should be recognized from its grain shape. See the international snow classification method for more detail (Fierz et al., 200). Consider using another word instead of "melt layer" throughout the manuscript.

P. 5, L. 132: Please indicate the position and altitude of the AWS.

Figure 1: It is better to put this figure in another figure showing the entire High Mountain Asia and India. At least, latitude and longitude should be indicated in the present figure. An explanation of dashed lines and accompanied numbers is also needed.

P. 5, L. 134 ∼ 135: "up" -> "upward", "down" -> "downward"

P. 5, L. 135 ∼ 136: Please indicate sensor types and manufactures not only for the snow depth sensor but also for all the instruments.

P. 5, L. 140 ∼ 141: "the incoming SW radiation is greater than outgoing SW radiation": This situation indicates the normal situation; the sentence should be revised.

P. 5, L. 142: The minimum snow albedo of 0.2 sounds extremely low to me. Please justify the value.

P. 6, L. 145: "100 kg m-3": Please justify the value.

P. 6, L. 155: Can the authors briefly explain the characteristics of the EUSAAR_2 protocol? Kuchiki et al. (2015) used the Interagency Monitoring of Protected Visual Environments (IMPROVE) thermal evolution protocol (Chow et al., 2001) to analyze snow samples with the Sunset OCEC analyzer. What is the key difference between

the EUSAAR_2 protocol and the IMPROVE protocol?

Figure 2: In my opinion, ECacc and SWEacc should be set in x-axis and y-axis, respectively. I believe such a figure may be intuitive for readers. Also, please consider changing colors for dashed lines and dashed-dotted lines; at present, it is a bit difficult to distinguish them from each other.

P. 7, L. 208 ∼ 209: The process to obtain the "suitable constants" (explained in the supplementary material) is not easy to follow. Because I assume this part may be an important part of this study, I recommend including this part in the main text. Also, adding figures showing scatter plots for different EC*constants may be nice.

P. 8, L. 245 ∼ 248: The assumption made here and the following discussion in this paragraph suggest that the authors consider most of the light-absorbing particles deposited through the wet deposition process in the study area. How about the dry deposition? Before suggesting the lateral transport (P. 9, L. 276 ∼ 278), the authors should consider the effects of dry deposition.

Figure 4: Like Figure 2, please consider setting equivalent precipitation and relative depth in x-axis and y-axis, respectively.

P. 10, L. 298 ∼ 307: So, what is the reason for the higher MD (mineral dust) fraction in the "melt layers"? Detailed discussion is needed.

P. 12, L. 376: "summers of 2015 and 2016": Do the authors mean that the melt layer formed as a result of the merge of the 2015 and 2016 summer melt layers? This point may not be explained in the Results and discussion section. Please explain more in detail at an appropriate place in the text.

References Chow, J. C., J. G. Watson, D. Crow, D. H. Lowenthal, and T. Merrifield (2001), Comparison of IMPROVE and NIOSH carbon measurements, Aerosol Sci. Technol., 34(1), 23–34.

Fierz, C., Armstrong, R. L., Durand, Y., Etchevers, P., Greene, E., McClung, D. M.,

[Figure]

Nishimura, K., Satyawali, P. K., and Sokratov, S. A.: The International Classification for Seasonal Snow on the Ground, IHP-VII Technical Documents in Hydrology N_83, IACS Contribution N_1, UNESCO-IHP, Paris, viii, 80 pp., 2009.

Kuchiki, K., T. Aoki, M. Niwano, S. Matoba, Y. Kodama, and K. Adachi, 2015: Elemental carbon, organic carbon, and dust concentrations in snow measured with thermal optical method and filter weighing: variations during 2007-2013 winters in Sapporo, Japan, J. Geophys. Res. Atmos., 120, 868-882, https://doi.org/10.1002/2014JD022144.

---

## Referee Comment (RC2) · Anonymous Referee #2 · 22 Dec 2020

This is a solid paper, and of significant value to the community interested in deposition of light-absorbing non-soluble materials in snow, especially in a region with huge societal implications for glacier water. I have only one comment that rises above "minor":

In supplemental material - the relationship between EC and TC is not well characterized as "tight correlation + random outliers", but rather looks like "two separate tight correlations". This indicates multiple populations of particulate with different EC:TC relationships, which is quite different than the single relationship that was assumed for the analysis (not only for estimation of EC in over-loaded filters, but also in terms of homogeneous behavior in different pits). I'd like to see this topic elevated to the main

text with some more information for the reader - are the "outliers" randomly distributed in the pits/depths, or is there some rhyme or reason to their distribution? If they are truly random, I agree that it's unlikely that the authors will be able to make use of them explicitly, but if not, they likely reveal some additional issues to be considered.

The rest of the comments are minor:

abstract - L25: please specify that they are within '2 km of each other L29: specify here if units are of liquid water equivalence or snow depth.

L 163 - pleas also specify the total number of filter samples. L 169 - Why is there an offset from zero? Are loadings high enough that this is negligible, or of no significance?

Figure 3- It seems more intuitive that the calculated ECacc (which is simply a scaling of liquid water depth) should be on the horizontal axis.

L239 : The main significance of the similarity of the ECacc to SWEacc seems to be in reference to the length-scales of variability in this region: they must be substantially longer than the $\sim$2 km separating the pits.

---

## Author Comment (AC1) · 22 Jan 2021

acp-2020-1059

Deposition of light-absorbing particles in glacier snow of the Sunderdhunga Valley, the southern forefront of Central Himalaya

Author's response

We thank both of the referees for taking the time to review our manuscript. Please find below our response (in *italics*) to the referees comments and added changes made in the revised manuscript in *red,* and text removed indicated with .

Anonymous Referee #1

General comments: This paper reports in-situ data for mass concentrations of light-absorbing particles (LAP) in snow such as black carbon, organic carbon, and dust collected at the Sunderdhunga Valley, the southern forefront of Central Himalaya during in October 2016. By comparing the data with in-situ surface meteorological data collected also by the authors, mechanisms that controlled the vertical redistribution of black carbon and dust within the near-surface snow are discussed. The methods to obtain these data are reliable. So, the data themselves are valuable for the community: e.g., the data tell us the current mass concentrations of LAPs within snow in the study area; and the data can be used for evaluation of chemical transport models. However, this reviewer would like to suggest that the authors should provide clear answers to the following major concerns before considering its publication:

1. The discussion on the redistribution of LAPs in "melt layers" is not enough to me. At least, the effects of dry deposition should be examined before considering the possibility of the lateral transport of LAPs in the near-surface snow.

*The melt layer represents many times the annual wet deposition of EC, which is why we consider the melt layer not to be a result of dry deposition. The exact mechanism is still unclear to us, but one possible explanation is some sort of lateral transport to account for the EC amounts encountered in the melt layers. A simple estimate on the dry deposition may help understand why we do consider it not very significant for the overall deposition. Using a dry deposition velocity of BC of 0.3 mm/s (Emerson et al., 2018) and an atmospheric concentration of 0.3 $\mu g\ m^{-3}$, reported at the Nepal Pyramid station during the pre-monsoon (i.e. season with high BC atmospheric concentration) (Bonasoni et al., 2010), the dry deposition can be estimated to contribute with 2838 $\mu g\ m^{-2}$ annually. In comparison to the BC wet deposition, which we can assume to be on the order of 40 000 $\mu g\ m^{-2}$ annually (obtained by multiplying our 50 $\mu g\ m^{-2}$ per $mm^{-1}$ with our annual precipitation estimate of 800 mm), it suggests that the BC dry deposition is about 7% of the total BC deposition.*

*In the revised manuscript this reasoning on the low contribution from dry deposition has been added towards the end of the first paragraph of section 3.2:*

*At the same time, the dry deposition of EC probably accounts for only a few percent of the deposition. With a dry deposition velocity of EC of 0.3 mm/s (Emerson et al., 2018) and an atmospheric*

*concentration of 0.3 µg m⁻³, reported at similar altitude at the Nepal Pyramid station during the pre-monsoon (Bonasoni et al., 2010), the dry deposition can be estimated to 2800 µg m⁻² annually, which several orders of magnitude lower than what is encountered in the enriched LAP layers. Thus, this leads us to propose…*

2. What is the main factor that induced higher concentration of MD (mineral dust) in the "melt layers"?

*We believe that the main factor that induces a higher concentration of MD in the melt layers is the same as EC. With melting of the surface layer MD becomes enriched at the surface—hence, being the same process for both EC and MD. If the referee is referring to the higher light absorbing fraction by MD for the melt layers compared to the other snow layers (highlighted in our fig. 5), this could be a result of the filter substrates being very dark for those specific snow samples (further explained in the second paragraph of section 3.3 in the manuscript).*

*With the relative enhancement (darkness of filter substrates) already explained in the text we have not made any changes to that section. The co-enhancement of both EC and MD with snow melt, on the other hand, we had not mentioned previously in the text. This is now mentioned at the end of first paragraph of the 3.3 section in the revised manuscript:*

*As with EC, MD has the propensity to remain at the snow surface with melting (e.g. Doherty et al., 2013).*

Specific comments

P. 1, L. 20: What do the authors mean by "scheme"? "Scheme" is often related to a sub-program of a numerical model. Therefore, using "state" may be better here.

*Thanks for pointing this out—'scheme' has now been replaced with 'state.'*

P. 1, L. 28: "similarities": Can the authors describe quantitatively?

*Similarities refers to the deposition that can be described by a constant for the different snow layers in the pits. The emphasis of the sentence has been changed in the revised manuscript to reflect this more accurately, now reading:*

*For the analyzed elemental carbon (EC), the  snow layers in the different pits show similarities in that the layers above and below the enriched LAP layer can be characterized by two constant  deposition values. Namely,  about 50 µg m⁻² per mm⁻¹ snow water equivalent (SWE) for young snow and 150 µg m⁻² per mm⁻¹ SWE for old snow. .*

P. 3, L. 43: "scheme": same as above.

*Corrected.*

P. 3, L. 57: "ppb" -> "ppbw" or "ppbv"?

*This refers to ppbw, which has now been adjusted.*

P. 3, L. 64 ~ 65: Observations are ground truth, so that it is not necessary for observations to be supported by modelling studies. On the other hand, observations can support the reliabilities of numerical models. Please reformulate this sentence.

*This is true, and the sentences has been reformulated to:*

* Recent modeling studies have reported analogous results, indicating certain sub-regions of the Himalaya to be especially vulnerable to LAP deposition.*

P. 5, L. 109: "hard layer": The "hardness" considered here may depend on the tools that the authors used to dig the pits. What kind of tools did the authors use to dig?

*It is certainly true that the tools used for digging will influence the penetrable pit depth. For these pits we used hardened hand shovels with a sharp edge—in order to be able to penetrate through small ice layers (compared to most traditional snow shovels which tend to not be very robust and thus, not very suitable to penetrate harder layers in snow pits). This information has been added in the revised manuscript:*

*The depth of the pits depended on the level at which a hard layer was found, and digging could not be further conducted with the reinforced shovels with a sharpened edge.*

P. 5, L. 115 ~ 117: Please show a figure showing profiles for snow density at these 2016 pits. Then, explain/indicate the layers where the dark, white, and gray snows can be found.

*We have added a figure in the supplement (fig. S1) with photographs of two typical pits from 2016 (pit B and D). The different snow layers are identified in this figure. Since density measurements could not be conducted below the LAP layer, we do not think there is a great interest in displaying them in the figure (or in any separate density profile). For densities we refer the reader to table 1 where they are presented.*

*Along with this newly added figure to the supplement, we added numerical section headers to the supplement.*

P. 5, L. 116 ~ 117: "a relatively thin (on the order of centimeters) very dark layer was separated by white snow above and more grey appearing snow below.": Is this sentence OK? Please check it again.

*We revised the sentence to: a relatively thin (on the order of centimeters) very dark layer was wedged in-between white snow above and more grey appearing snow below.*

P. 5, L. 118: Strictly speaking, "melt layer" should be recognized from its grain shape. See the international snow classification method for more detail (Fierz et al., 2009). Consider using another word instead of "melt layer" throughout the manuscript.

*This is a good observation by the referee. We have changed 'melt layer' to 'enriched LAP layer' throughout the manuscript.*

P. 5, L. 132: Please indicate the position and altitude of the AWS. Figure 1: It is better to put this figure in another figure showing the entire High Mountain Asia and India. At least, latitude and longitude should be indicated in the present figure. An explanation of dashed lines and accompanied numbers is also needed.

*The AWS location is indicated with a star in Fig. 1. We have now added the altitude of the AWS next to the star-marker; latitude and longitude to the fig., as well as clarified in the figure text that the dashed lines and numbers inside the glacier outlines refer to contour lines. For an overview picture we refer to Svensson et al., (2018) where this was already published. The figure text now reads:*

*Figure 1. Map of glaciers with the location of the snow pits (black dots) and AWS indicated* with a star (and altitude). Dashed lines inside the glacier outlines refer to altitudinal contour lines.

P. 5, L. 134 ~ 135: "up" -> "upward", "down" -> "downward"

*Corrected.*

P. 5, L. 135 ~ 136: Please indicate sensor types and manufactures not only for the snow depth sensor but also for all the instruments.

*Sensor specifics have been added for all AWS instruments.*

P. 5, L. 140 ~ 141: "the incoming SW radiation is greater than outgoing SW radiation": This situation indicates the normal situation; the sentence should be revised.

*This does in-deed indicate the normal circumstances, where the 'incoming SW is greater than outgoing SW radiation.' For a limited amount of data this was not the case, and we believe it is due to the upward sensor being covered by snow, generating cases where this was the case. It is for this reason that we use the logical conditions to remove this data. With that, we do not see how the sentence should be revised.*

P. 5, L. 142: The minimum snow albedo of 0.2 sounds extremely low to me. Please justify the value.

*A snow albedo of 0.2 is in fact very low. We would like to point out that this was used to filter the data and during the times when this value was obtained the snow height was also very low. The albedo of the ground therefore influences this albedo value (if not the main dominating parameter influencing the AWS snow albedo during those times). The bare ground at the AWS site had an albedo of 0.17. It is due to this reason that we used 0.2 as a screening parameter to our albedo data. In the revised manuscript we clarified this data filtering approach:*

*'to ensure snow cover* as the ground albedo was measured at 0.17.'

P. 6, L. 145: "100 kg m-3": Please justify the value.

*We chose to work with 100 km m$^{-3}$ partly for simplicity, and the fact that it is common to work with that value for new snow. Admittedly however, we are aware of the variation in the density of new snow that exists. For the revised manuscript we added a reference (Helfricht, et al., 2018), to support our selected value of 100.*

P. 6, L. 155: Can the authors briefly explain the characteristics of the EUSAAR_2 protocol? Kuchiki et al. (2015) used the Interagency Monitoring of Protected Visual Environments (IMPROVE) thermal evolution protocol (Chow et al., 2001) to analyze snow samples with the Sunset OCEC analyzer. What is the key difference between the EUSAAR_2 protocol and the IMPROVE protocol?

*The EUSAAR_2 is a standardized protocol within the EU countries that was developed in 2010 to minimize artefacts in OC/EC analysis (e.g. charring and carbonate interference) (Cavalli et al., 2010). To our knowledge IMPROVE is frequently associated with DRI-analyzer and not the Sunset OCEC analyzer*

*(but IMPROVE protocol can easily be run on the Sunset instrument on the other hand). The main difference between EUSAAR_2 and IMPROVE is in the number of temperature steps, with EUSAAR_2 containing an additional step during the EC phase; and the temperature reached at each stage are different; as well as the duration at each step. To our knowledge the different protocols have not been evaluated for EC measurements in snow, but for airborne EC similar results have been reported when comparing the different protocols (e.g. Bautista et al., 2015).*

Figure 2: In my opinion, ECacc and SWEacc should be set in x-axis and y-axis, respectively. I believe such a figure may be intuitive for readers. Also, please consider changing colors for dashed lines and dashed-dotted lines; at present, it is a bit difficult to distinguish them from each other.

*We switched the axes for ECacc and SWEacc. We want the lines to not be the focus of the figure, but rather the black data points, which is why we would like to keep the lines in grey in order to be noticeable (but not overtaking the figure). We switched the dash-dotted line (ECy) to dotted line in hopes of distinguish it more from the dashed line.*

P. 7, L. 208 ~ 209: The process to obtain the "suitable constants" (explained in the supplementary material) is not easy to follow. Because I assume this part may be an important part of this study, I recommend including this part in the main text. Also, adding figures showing scatter plots for different EC*constants may be nice.

*The important part that we want to highlight is the homogeneity between each pits different snow layers (i.e. young and old snow), and that we are able to display that with simple common deposition constants. The process of how those constants were determined we believe is not essential to the main text. With figure S3 (now fig. S4) we have in way already tested a range of scatter plots with different EC*constants (for both young and old snow with the different lines) and their resulting slopes. We believe condensing this sensitivity to the different constants with this figure is more valuable than presenting different scatter plots for different EC*constants (which all would show similar results when constants are around 50 and 150 for young and old snow, respectively). In the revised supplement we revised this section's text to make it easier to follow.*

4.  *Determining  suitable common effective constants for young and old snow ($EC_y^*$ and $EC_o^*$), respectively.*

*In order to define the constants $EC_y^*$ and $EC_o^*$ we systematically changed the constants over a range of values and plotted (in Fig. S4) the returned slope from the linear fit between observed ECacc and calculated ECacc . Where the linear fit returns a slope of 1, the ideal EC*constant is found.. Evident in Fig. S4, the precise $EC_y^*$ constant is slightly more than 50 $\mu g\ L^{-1}$, while the $EC_o^*$ constant is somewhat lower than 150 $\mu g\ L^{-1}$. For convenience and simplicity, however, we chose to work with the numbers 50 and 150 for $EC_y^*$ and $EC_o^*$, respectively.*

P. 8, L. 245 ~ 248: The assumption made here and the following discussion in this paragraph suggest that the authors consider most of the light-absorbing particles deposited through the wet deposition process in the study area. How about the dry deposition? Before suggesting the lateral transport (P. 9, L. 276 ~ 278), the authors should consider the effects of dry deposition.

*This issue has been addressed in the first major comment (above).*

Figure 4: Like Figure 2, please consider setting equivalent precipitation and relative depth in x-axis and y-axis, respectively.

*This has been changed in the revised manuscript.*

P. 10, L. 298 ~ 307: So, what is the reason for the higher MD (mineral dust) fraction in the "melt layers"?

*This issue has been addressed in the second major comment above.*

P. 12, L. 376: "summers of 2015 and 2016": Do the authors mean that the melt layer formed as a result of the merge of the 2015 and 2016 summer melt layers? This point may not be explained in the Results and discussion section. Please explain more in detail at an appropriate place in the text.

*Yes, we are arguing that the melt layer is formed as a result of the merge of the 2015 and 2016 summer layers. But, as we attempt to discuss towards the end of section 3.2, we further argue that the melt layer is not only a result of the merging of these two summer melt layers. With the amount of EC in the melt layer, we do not see that the merging of the two summer layers explain the observed EC amounts, but also that EC was accumulated in another way—leading us to propose the convergence of material surrounding the sampling pits. To clarify this further we added in the revised manuscript:*

*'This layer was most likely a result of strong melting that took place in the summers of 2015 and 2016 as discussed in 3.2.'*

*Furthermore, we noticed a mistake in the abstract on this topic (lines 32-33), which we revised to the following:*

*The  enriched LAP layer is likely a result of strong melting that took place during the summers of 2015 and 2016, as well as possible lateral transport of LAP.*

Anonymous Referee #2

This is a solid paper, and of significant value to the community interested in deposition of light-absorbing non-soluble materials in snow, especially in a region with huge societal implications for glacier water. I have only one comment that rises above "minor":

In supplemental material - the relationship between EC and TC is not well characterized as "tight correlation + random outliers", but rather looks like "two separate tight correlations". This indicates multiple populations of particulate with different EC:TC relationships, which is quite different than the single relationship that was assumed for the analysis (not only for estimation of EC in over-loaded filters, but also in terms of homogeneous behavior in different pits). I'd like to see this topic elevated to the main text with some more information for the reader - are the "outliers" randomly distributed in the pits/depths, or is there some rhyme or reason to their distribution? If they are truly random, I agree that it's unlikely that the authors will be able to make use of them explicitly, but if not, they likely reveal some additional issues to be considered.

*After reviewing the text in the supplement on this topic we can see that this was not well explained and we did not screen the data consistently before determining the linear fit used for reconstructing EC values for saturated samples. Our point is that with higher TC loading (which coincides with filter samples where there is an overall heavy LAP loading) the EC values tend to be underestimated, and we want to adjust these EC values. Consistent for the data points is that the 'outliers' are from a snow sample layer that contains a higher LAP amount (i.e. melt layer sample or old snow sample layer), and they are present in most of the different pits. From that view, the 'outliers' are randomly placed.*

*Thanks to the referee's comments on this topic we used a more stringent selection for which data points that are included for generating the relation between TC and EC, namely not exceeding TC 100 µg cm$^{-2}$ and an optical depth of less than two (measured with the PSAP prior to OCEC-analysis). We added the last step since with a high overall LAP loading on the substrates non-linear effects come into play. Using these data screening criterion it resulted in the new supplement figure S3a, where all of the filtered data points are presented together with one slope (0.10) and all the data points prior to screening with another slope (0.027). Note that the different year's samples are marked differently in the figure and are further highlighted in Fig. S3b (displays an excerpt from fig. S3a). Due to the comment on the offset (minor comment by the referee below), we force the slope through origo and ignore the offset since it will be negligible for the samples where EC is reconstructed. In the end the slope remains identical 0.1 vs. 0.1 used in the previous version of the manuscript, but nonetheless, shifted a few data points that were included or excluded in the relation. Overall, the EC results were marginally adjusted. In the revised manuscript the numbers have been updated. As we do not consider this issue to be in needed to be elevated to the main text, since it would redirect the readers focus, we believe in keeping these figures (+text) in the supplement. In the revised manuscript we did, however, update the main text with:*

*A slope of 0.1 for the EC:TC ratio for filter samples considered non-saturated*  *was used to reconstruct the EC content for the filter samples containing high amounts of absorbing particles (see details in supplement and Fig. S3a-b).*

*And the supplement text (and figures) we changed to:*

1. *Determining TC/EC*  *ratio for unsaturated filter samples*

~~First, a plot of EC as a function of TC containing the 2016 filter samples where the EC amounts were assumed to be reliable (i.e. excluding the 17 samples where accurate EC determinations could not be done) is presented in S23a with the slope of 0.023. In this plot it was evident that some outliers were present in the data set, indicated by triangles in S23a. With these outliers removed in Fig. S23a, we obtained the slope 0.13 (for all black squares). Using a similar approach for the 2015 filter samples (S23b), we first obtained a slope of 0.022, and then a plot where samples that had a TC amount that was higher than 70 μg was removed, yielding a slope of 0.088. Grouping the filtered data points from 2015 and 2016 together, we obtained the linear fit EC = 0.10TC + 0.12. This linear fit was applied to the samples where EC was reconstructed.~~

*When plotting filter samples where the EC amounts were assumed to be reliable (i.e. excluding the 17 samples where accurate EC determinations could not be done), it became evident that some data points were unsatisfactory—containing elevated LAP loadings, lowering the EC:TC ratio (Fig. S3a). Therefore, the data was excluded according to: TC lower than 100 μg cm$^{-2}$, as well as an optical depth of less than two. With these data points removed (N=11), the EC:TC resulted in a higher slope (0.1;Fig. S3a). In Figure S3b the filtered data points are shown, demonstrating that the slope is nearly identical between the two sampling years. Since the offset will be negligible for the samples where EC is reconstructed, it was here ignored. The 17 samples where accurate EC determinations could not be done were reconstructed with the slope of 0.1.*

[Figure]

*Figure S**3. EC TC ratio for (a)*  *All filter samples, as well as filtered data points; and (b)*  *only filtered data points.*

The rest of the comments are minor:

abstract - L25: please specify that they are within '2 km of each other L29: specify here if units are of liquid water equivalence or snow depth.

*In the revised manuscript we specify the pits distance with one-another (one km); and that the units refer to snow water equivalent.*

L 163 - please also specify the total number of filter samples. L 169 - Why is there an offset from zero? Are loadings high enough that this is negligible, or of no significance?

*We have now added the total number of filter samples (91). The referee is correct; for the filter samples where the EC is reconstructed the TC loading is so high that the offset is negligible (on the order of a few percent), thus we do not take into account the offset.*

Figure 3- It seems more intuitive that the calculated ECacc (which is simply a scaling of liquid water depth) should be on the horizontal axis.

*In the revised manuscript the axes have been changed.*

L239 : The main significance of the similarity of the ECacc to SWEacc seems to be in reference to the length-scales of variability in this region: they must be substantially longer than the ~2 km separating the pits.

*It is not clear to us what the referee is referring to, and whether the referee expects similarities or not within the kilometer scale for the pits. Since we are not able to anticipate similarities or not prior to when the pits were sampled and analyzed, we do find this result interesting. In addition, the similarities between the sampling years adds to the fact that we do find this result exciting.*

References

Bautista et al. (2015) doi:10.5094/APR.2015.037.

Bonasoni et al. (2010) doi:10.5194/acp-10-7515-2010

Cavalli et al. (2010) doi.org/10.5194/amt-3-79-2010, 2010.

Emerson et al. (2018) doi.org/10.1029/2018JD028954.

Helfricht et al. (2018) doi.org/10.5194/hess-22-2655-2018.